# Elaboration and Characterization of Dulce de Leche with Reduced Sugar Content

Victor F. Moebus [1,*], Leonardo A. Pinto [2], Felipe B. N. Köptcke [3] and Luiz A. M. Keller [4]

1 Food Technology Department, Veterinary Medicine School, Universidade Federal Fluminense, Niterói 24020-141, RJ, Brazil
2 Health Sciences Center, Universidade Federal do Rio de Janeiro, Rio de Janeiro 21941-909, RJ, Brazil
3 Pharmacy School, Universidade Federal Fluminense, Niterói 24210-346, RJ, Brazil; felipekoptcke@id.uff.br
4 Department of Animal Husbandry and Sustainable Agro-Social-Environmental Development, Veterinary Medicine School, Universidade Federal Fluminense, Niterói 24210-346, RJ, Brazil; luiz_keller@id.uff.br
* Correspondence: victormoebus@id.uff.br; Tel.: +55-21985822408

**Abstract:** Dulce de leche is a South American traditional dairy product. However, due to the high sugar content, it is unsafe for consumers with special physiological conditions, such as diabetes. Light foods were developed to meet those demands whilst maintaining the sensory characteristics. The present study aimed to develop and characterize a formulation of light dulce de leche, using sweeteners to substitute sucrose, and gums as thickening and stabilizing agents. The physicochemical assays follow the AOAC manual, and the microbiological parameters were set according to MERCOSUL's legislation. Thirty potential consumers perform the nine-point hedonic scale for product acceptance and the seven-point scale for buying intention. Compared to the traditional formulation, the product had better rheological properties, but a lighter color. All the raw materials and final products were considered safe by the recommended microbiological and toxicological standards; however, the product showed discrepancies to the physicochemical requirements. The formulation had an overall medium acceptance and low buying intention. The product had deficiencies, needing other compounds to reach the desired characteristics. Even though it leads to a more expensive final product, it is possible to adjust the product and allow access to more selective consumers or with restrictions.

**Keywords:** nutraceuticals; light foods; food safety





## 1. Introduction

Dulce de leche is a traditional dairy product of South American countries, especially Argentina and Brazil. Dulce de leche can be defined as the product obtained by heat action of concentrated or reconstituted milk. It may include the addition of other food substances, leading to a variety of final products traditionally consumed as a dessert or ingredient for pastry, cakes, biscuits, and ice cream [1,2].

As for the technological process, the dulce de leche can be made at normal or reduced pressure and with or without the addition of solids of milk origin and/or cream of sucrose. The addition of sucrose (partially replaced or not by monosaccharides and/or other disaccharides) has been used to increase the final product and ensure adequate total solids concentration without affecting the moisture of the final product [2].

In recent years, however, consumers' searches for healthy foods has gained prominence due to the growing concern with food-related diseases, with many opting for foods with total or partial reduction of certain components, such as sugars. Diet and light foods were developed as a response to meet the demand of consumers with special physiological conditions, such as hypertension, diabetes, and obesity; however, such products are also consumed by individuals who wish for a more balanced and healthy diet, but without giving up the sensory characteristics [3–5].

The Brazilian legislation requires a reduction of at least 25% in energy value or nutrient content in relation to the reference food for the product to be considered reduced or light [6]. Brazilian legislation also establishes a limit of 40% of sucrose concentration in dulce de leche [2]. Those legal demands, associated with consumer demand, encourage the dulce de leche industries to reduce the addition of sucrose and add gums and sweeteners in a way that the energy value and nutrient content are reduced, but the characteristics and flavor of the product are maintained [7].

Sweeteners can be used as a substitute for sucrose, not providing energy when metabolized by the human organism, being classified as non-nutritive. Among the sweeteners used by the food industry, sucralose must be highlighted, having a sweetening power 400 to 800 times greater than sucrose. Gums are polymeric compounds that form viscous solutions or dispersions when dissolved or dispersed in water, having wide applications as thickening and stabilizing agents [7].

In this context, the present study aimed to develop a dulce de leche light with sugar reduction and determine its nutritional composition and its physicochemical and microbiological characteristics in order to compare to those established in the legislation.

## 2. Materials and Methods

### 2.1. Elaboration of Dulce de Leche Formulations

Two formulations of dulce de leche were developed, traditional dulce de leche used as a control group, and light dulce de leche with sugar reduction.

Pasteurized milk and other ingredients were purchased from local commerce and the product was developed in the Physical-Chemical Control laboratory of the Food Technology Department at the Fluminense Federal University. The preparation of the formulations was in an industrial pan with a double jacket and mechanics using the raw materials enlisted in Table 1 below:

**Table 1.** Dulce de Leche Formulation's Raw Materials.

| Traditional Dulce de Leche | | Light Dulce de Leche | |
|---|---|---|---|
| Pasteurized Milk | 3.0 L | Pasteurized Milk | 3.0 L |
| Sucrose | 500.0 g | Sucrose | 250.0 g |
| Glucose syrup | 100.0 g | Glucose syrup | 50.0 g |
| Baking soda | 2.4 g | Baking soda | 2.4 g |
| - | - | Sucralose | 30.0 g |
| - | - | Whole milk powder | 279.0 g |
| - | - | Carboxymethylcellulose | 10.84 g |
| - | - | Xanthan gum | 0.51 g |
| - | - | Guar gum | 0.51 g |
| - | - | Caramel coloring (INS 150[a]) | *q.s.p.* |

*q.s.p.*: Quantum satis para.

For the traditional dulce de leche, pasteurized milk was added to the pan with acidity correction by baking soda and heated to 110 °C. After boiling for 30 min, the other ingredients were added, while heating and stirring until reaching the desired point (65–68° Brix).

The production of light dulce de leche proceeded as the control. For this, 2 L of pasteurized milk were boiled for 30 min at 110 °C. The other ingredients were previously solubilized in the remaining milk to avoid the formation of lumps and then added under slow stirring and heating until 65–68° Brix.

The evaluation of the ideal point was accomplished by dripping the hot product into cold water during the processing and continuous investigation of the formation of spherical and solid droplets of dulce de leche.

The Brix degrees were measured using a Compact Brix digital refractometer (AKSO®, São Leopoldo, RS, Brazil). After reaching the desired Brix degree, the formulations were cooled to 70 °C and filled in previously sanitized plastic containers.

## 2.2. Physical-Chemical and Nutritional Characterization

Physicochemical assays were done in triplicate following Brazilian Ministry of Agriculture, Livestock and Supply (MAPA) guidelines [2,8,9] based on the Official methods of analysis of the Association of Official Analytical Chemistry (AOAC) [10].

Samples were dried at 105 °C to obtain humidity and total solids content. From the dried samples, the samples were subjected to calcination to determine the total mineral material. The total crude protein content was determined by indirect determination through a distillation method. Total lipid content was obtained through the Rose-Gottlieb method as described in the AOAC manual [10].

Rheological analysis was performed to evaluate the similarity between the two products developed. The rheological analysis was covered by texture profile analysis (TPA, gumminess, hardness, adhesiveness, and cohesiveness) being performed using a TA-XT plus texture analyzer (Stable Micro Systems, Godalming, UK), according to Leddomado et al. [11]. Then, 250 g of each sample were compressed 10 mm using the following parameters: pretest speed of 2 mm s$^{-1}$, test speed of 1 mm s$^{-1}$, posttest speed of 2 mm s$^{-1}$, time of 0.5 s, and contact force of 1 N.

The total energy value of the developed dulce de leche formulation was calculated based on the conversion factors for proteins (4 kcal g$^{-1}$), carbohydrates (4 kcal g$^{-1}$), and lipids (9 kcal g$^{-1}$). The data obtained in kcal were converted to kJ using the factor: 1 kcal = 4.184 kJ.

## 2.3. Microbiological and Toxicological Characterization

Microbiological parameters were performed following Brazil's legislation for dulce de leche, fluid milk, and powdered milk, analyzing both the raw material and the final product. To assess the security of the product and the quality of handling, the presence of coagulase-positive *Staphylococci*; total mesophilic bacteria; Coliforms; *Salmonella* spp.; filamentous fungi; and yeast were evaluated [12]. Total mesophilic bacteria counts were carried out to evaluate the raw material and evaluate manufacturing quality. Analyses were performed as described in the Compendium of Methods for the Microbiological Examination of Foods [13].

From 25 g of sample, serial dilutions were made up to a ratio of 1:1000. Each dilution was inoculated into Petri dishes containing specific culture media for each microorganism studied. Viable cell counts after incubation were performed through plate counts using a Biocell colony counter model Biocc150-Bi (Prolab$^{®}$, São Paulo, Brazil). After the first evaluation, microbiologic analyses were repeated after 15 and 30 days of preparation to verify the quality of the product subjected to adequate storage and stipulate its commercial validity.

Toxicological evaluations were realized considering the health risks associated with the presence of these substances. The staphylococcal enterotoxin was evaluated as required by legislation. The screening for specific genes responsible for staphylococcal enterotoxin was performed as described by Pyzik et al. [14]. Aflatoxin M1 (AFM1) detection and quantification were performed in raw materials following the description by Moebus et al. [15].

## 2.4. Product Acceptance

The acceptance test of the prepared products was carried out after checking the microbiological quality of the formulations. Approximately 30 potential consumers among university students were invited.

The dulce de leche samples were presented to consumers in portions of approximately 30 g, in 50 mL plastic cups, identified by random numbers. The samples were served at room temperature and, during the execution of the tests, water and unsalted cream crackers were made available for the participants to clean the oral cavity between tasting each sample [16].

The participants answered questionnaires afterward about the assessment of global preference and purchase intention for the dulce de leche samples. It used an acceptance test with a 9-point structured hedonic scale for the preference with its extremes being 1,

meaning "liked it very much" and 9, meaning "I disliked very much". A structured 7-point scale was used for the buying intention with its extremes being "definitely would not buy" and "definitely would buy" [17].

*2.5. Statistical*

Results are expressed as the means and standard error means (SEM) of three experiments with triplicate determinations. All statistical analyses were performed using the Infostat version 2020 software (Infostat Software, Córdoba, Argentina). Statistical treatments were carried out by comparing means and frequencies employing the Tukey tests and frequency assessments (ANOVA) at a significance level of 5% ($p \leq 0.05$) for treatment and experimental group comparisons [18].

### 3. Results and Discussion

*3.1. Dulce de Leche Characterization*

The production of the traditional dulce de leche formulations took about 3 h to obtain the desired ideal point with a yield of 34.0%, obtaining 1.020 Kg from 3 L of pasteurized milk. The total solids concentration of the final product was 69.7° Brix, corroborating the desired endpoint for the product.

The traditional dulce de leche presented the characteristic color and odor of the product, obtained through the caramelization of carbohydrates and the Maillard reaction [19]. The consistency of the final product was also characteristic of the product. The concentration of solids through the evaporation of water results in an increase in the viscosity of the formulation, providing the desired rheological characteristics [20].

The light dulce de leche formulation also took about 3 h to obtain the desired point with a 32.9% yield, achieving 987 g from 3 L of pasteurized milk. The time and productivity of the formulations were similar, in a way that the productive difference between them was not significant.

The light dulce de leche reached 56.6° Brix, which is an acceptable value considering that there must be a reduction in the nutrient content for light food. Due to the reduction of total sugar, however, the characteristic color of the Maillard reaction did not occur in this formulation, leading to a lighter color product and requiring the addition of caramel coloring to adjust until a desirable characteristic [21].

The texture of the finished product was thick only due to the food additives, as the concentration of total solids with sugar reduction does not increase the viscosity in the final product. The use of milk powder and thickeners to compose the rheological characteristics of the product are important methods, making it necessary, however, to process the final dough in a mixer to give the final product creaminess [20].

The sucralose used in this work followed the established by the manufacturer's information, where the sweetening power is equivalent to 100 times the conventional sucrose. So, 25 g of sucralose was used to result in the desired concentration, and another 5 g of sucralose was added at the end to enhance the sweet taste. The addition was carried out at the end of the process to avoid prolonged contact with heat that could cause the sucralose's degradation [22].

The various sucralose on the market has different sweetening potential. For the development of formulations and to maintain the approximate flavor, it is necessary to take this potential into account and adjust the recipes accordingly.

While there is a considerable reduction in the amount of sucrose present in the mixture, for the light dulce de leche to present a similar characterization to the traditional one, it is necessary to add other compounds. This can lead to a more expensive final product. However, it also allows more selective or restricted consumers to access the product.

*3.2. Physical-Chemical and Nutritional Characterization*

After carrying out the aforementioned analysis tests, the physical-chemical data were arranged in Table 2 below with the principal nutrient composition.

**Table 2.** Centesimal Composition and Legislation.

|  | Moisture | Ashes | Lipids | Proteins |
|---|---|---|---|---|
| MERCOSUL Legislation * | <30.00% | <2.00% | 6.00–9.00% | >5.00% |
| Traditional Dulce de Leche | 29.88 ± 0.93% | 1.42 ± 0.11% | 2.65 ± 0.23% | 7.54 ± 0.55% |
| Light Dulce de Leche | 35.63 ± 1.60% | 2.88 ± 0.02% | 3.50 ± 0.33% | 12.07 ± 0.19% |

* MERCOSUL identity and quality standard [23].

The dulce de leche parameters follow the MERCOSUL legislation [23]. This guideline standardizes the criteria of the economic block formed by most countries in South America. Within these guidelines, dulce de leche has a series of standards, as described in Table 2.

Based on these data, we can see that the traditional dulce de leche moisture (29.88%) and ashes (1.42%) remained within the legislation (<30.00%; <2.00%), whereas the light dulce de leche (35.63% and 2.88%, respectively) surpassed both established limits. Both formulations had an amount of fat below the established by MERCOSUL (6.00–9.00%), with the traditional dulce de leche reaching a value of 2.65% whilst the light formulation had a value of 3.50%. Both also showed a protein percentage above the minimum required, as the traditional showed 7.54% and the light, 12.07%.

The moisture difference observed is probably caused by the thickening additives that must have evaporated along with the water. The differences observed in the ashes, lipid concentration, and protein percentage may be associated with the addition of powdered milk, since it has its solids and is rich in protein and fat. This difference was expected and proves the direct influence of powdered milk that significantly increases the value of the nutrient in the light formulation [24].

When evaluating the rheological parameters of the developed dulce de leche, both samples showed excellent adhesiveness and firmness, with the gumminess of the light formulation being slightly higher but with no significant difference observed ($p < 0.05$). The results obtained demonstrate that, qualitatively, the replacement of sugar with the other ingredients can offer the necessary characteristics for the characterization of the product. Higher gumminess, adhesiveness, and firmness are undesirable characteristics in DL used for cooking or confectionery purposes. In the case of direct consumption, the increase in texture parameters is well accepted by consumers [1].

Regarding the energy value, the value for traditional dulce de leche was 288.05 Kcal, and for the light product 263.46 Kcal. According to Brazilian legislation [6], for a product to be considered "light", the reduction in energy value should be at least 25.0%. The observed values represent a reduction of 8.54%, and, therefore, the light formulation was not successful in this regard. The limited reduction in energy content can be elucidated by the incorporation of components aimed at substituting sucrose in the formulation, notably powdered milk, which has the potential to elevate protein, fat, and lactose levels [25]. Notwithstanding the presence of lactose originating from whole milk powder, the reduction in additional glucose content was adequate to ensure that the product remains in accordance with the regulatory guidelines pertaining to added sugars in dulce de leche.

However, although the desired caloric reduction was not obtained, future studies focused on improving the methodology used to obtain products with functional characteristics since the comparative results between the two samples proved to be favorable.

*3.3. Microbiological and Toxicological Characterizations*

The microbiological parameters of the two formulations of dulce de leche and their raw material can be observed in Table 3 below:

**Table 3.** Microbiological characterization.

| Microorganisms | Brazilian Legislation * | Fluid Milk | Powdered Milk | Traditional Dulce de Leche | Light Dulce de Leche |
|---|---|---|---|---|---|
| Total mesophilic bacteria [1] | - | $4.08 \pm 0.08$ | $2.40 \pm 2.40$ | <1.0 | <1.0 |
| Coagulase-positive *Staphylococcus* [1] | 2.0 | <1.0 | $2.05 \pm 2.05$ | <1.0 | <1.0 |
| Total coliforms [1] | 2.0 | <1.0 | <1.0 | <1.0 | <1.0 |
| Thermotolerant coliforms [1] | 2.0 | <1.0 | <1.0 | <1.0 | <1.0 |
| Filamentous fungi and yeast [1] | 2.0 | $2.08 \pm 0.12$ | $3.15 \pm 3.05$ | <1.0 | <1.0 |
| *Salmonella* spp. [1] | Absent | Absent | Absent | Absent | Absent |

* Microbiological standards according to Brazilian legislation [12]; [1]: CFU counts expressed in $\log_{10}$ CFU $g^{-1}$; Limit of detection (LOD) $\leq 1.0 \log_{10}$ CFU $g^{-1}$.

Regarding the safety parameters, despite the risk of post-processing contamination, all the raw materials and both final products were considered safe to consume according to recommended microbiological standards.

The appliance of a quality standard to the raw materials for human consumption products is needed to ensure final product safety. The selection of high-quality materials is an important factor in ensuring the standard and security of the final product, as a higher contaminant concentration can be a risk to the whole manufacturing and the consumers, especially in highly manipulated products [2,23].

The microbiological concentration in the raw materials, while below the legislation limit [12], has a significant difference in comparison to both dulce de leches formulations. That significant decrease in microbial populations indicates that the manufacturing process may contribute to the conservation and security of the final products. Heat treatment, as well as high sugar concentration, are traditional procedures to reduce the microbiological population [1,26], and their association can be observed in both formulations.

The total mesophilic counts observed in this work confirm the efficiency of the heat treatment applied. In the raw material, total mesophilic counts were above $2.40 \log_{10}$ CFU $g^{-1}$ and drastically reduced after thermal processing, being present in minimal concentrations in the final products. Although not required in Brazilian legislation, the total mesophilic bacteria count is an important parameter to be evaluated during the development of a product, being directly correlated with the sanitation of the process [27–29].

The microbiological evaluation after the storage period is presented in Table 4:

**Table 4.** Microbiological evaluation after refrigerated storage.

| Microorganisms | Brazilian Legislation * | 15 Days Storage | | 30 Days Storage | |
|---|---|---|---|---|---|
| | | Traditional Dulce de Leche | Light Dulce de Leche | Traditional Dulce de Leche | Light Dulce de LECHE |
| Total mesophilic bacteria [1] | - | <1.0 | <1.0 | <1.0 | <1.0 |
| Coagulase-positive *Staphylococcus* [1] | 2.0 | <1.0 | <1.0 | <1.0 | <1.0 |
| Total coliforms [1] | 2.0 | <1.0 | <1.0 | <1.0 | <1.0 |
| Thermotolerant coliforms [1] | 2.0 | <1.0 | <1.0 | <1.0 | <1.0 |
| Filamentous fungi and yeast [1] | 2.0 | <1.0 | <1.0 | <1.0 | <1.0 |
| *Salmonella* spp. [1] | Absent | Absent | Absent | Absent | Absent |

* Microbiological standards according to Brazilian legislation [12]; [1]: CFU counts expressed in $\log_{10}$ CFU $g^{-1}$; Limit of detection (LOD) $\leq 1.0 \log_{10}$ CFU $g^{-1}$.

As seen in the table above, after a 30 day storage period under refrigeration, it was not possible to observe microbial growth in the evaluated formulations, attesting that the products remained suitable for consumption under adequate storage.

The use of heat treatment and reduction of water content as a food preservation technique aims to inactivate and prevent microbial growth with the maintenance of the parameters indicating that effectiveness. The treatment shows the capacity to inactivate the

initially viable cells, prolonging their storage period and serving as a basis for stipulating shelf life for food products [30–32].

The application of heat for this purpose, however, presents risks to the quality of the product, which may cause changes in the sensorial and nutritional characteristics of the product. That also influences the determination of the product's shelf life [33].

Regarding the antimicrobial properties of higher sugar concentration, it is known that hypertonic media can generate cell lysis in different types of bacteria, and those resistant to this physical phenomenon enter a state of latency. [34,35]. Both dulce de leche formulations present this effect, especially after the reduction of water content during heat treatment, resulting in a high sugar concentration. The moisture reduction itself also generates an inhibitory effect in most microorganisms [36].

In this way, the addition of sugar and sweeteners not only has the role of technologically assisting the processing of dulce de leche, but also plays an important role in the conservation of the final product when associated with the thermal treatment, guaranteeing a safe product for consumption and with a higher shelf life.

As for the toxicological evaluations, none of the evaluated samples showed a positive result for staphylococcal enterotoxin and AFM1, in compliance with Brazilian legislation.

Staphylococcal enterotoxins are primarily responsible for food poisoning caused by *Staphylococcus aureus*, often found in milk and dairy products [36,37]. Pasteurization effectively reduces the concentration of microorganisms in milk and, consequently, in its derivatives. Despite this, the detection of staphylococcal enterotoxins is necessary as it is thermoresistant and can cause outbreaks of food poisoning even in the absence of the pathogen [37,38].

While the Brazilian legislation does not present maximum limits allowed for AFM1 in dulce de leche, the quantification of raw materials and final products, especially dairies, is essential to guarantee the consumers' security and good health. AFM1 is a thermoresistant substance with global occurrence, having carcinogenic, cytotoxic, teratogenic, and genotoxic properties, and prolonged exposure can also cause chronic conditions such as immunosuppression, hepatocarcinoma, and stunted growth in children [39–41].

### 3.4. Product Acceptance

From the analysis of the data acquired from the questionnaires, it could be stipulated the average values for product acceptance, as well as flavor, texture, and appearance perception. The data were arranged in the following table (Table 5).

**Table 5.** Acceptance percentage for traditional dulce de leche.

| Score | Acceptance | Flavor | Texture | Appearance |
|---|---|---|---|---|
| 1 | 26.7% | 33.3% | 33.3% | 30.0% |
| 2 | 46.7% | 36.7% | 30.0% | 33.3% |
| 3 | 23.3% | 26.7% | 23.3% | 30.0% |
| 4 | - | - | 3.3% | 3.3% |
| 5 | 3.3% | 3.3% | 3.3% | - |
| 6 | - | - | 3.3% | 3.3% |
| 7 | - | - | 3.3% | - |
| 8 | - | - | - | - |
| 9 | - | - | - | - |
| Average Score | $2.1 \pm 0.89$ | $2.0 \pm 0.95$ | $2.4 \pm 1.49$ | $2.2 \pm 1.11$ |

According to the average values found, the traditional dulce de leche obtained a value close to 2.0 in all the requirements, meaning an acceptance degree of "I like it very much", which indicates approval from the participants.

The light formulation, as seen in Table 6 below, obtained an average score of 4.0 in almost all aspects, except for appearance, which was 4.6. The results indicate an acceptance level close to the middle of the scale, in the range of "I liked it slightly".

**Table 6.** Acceptance percentage for light dulce de leche.

| Score | Acceptance | Flavor | Texture | Appearance |
|:---:|:---:|:---:|:---:|:---:|
| 1 | 6.7% | 10.0% | 10.0% | 13.3% |
| 2 | 6.7% | 13.3% | 13.3% | 3.3% |
| 3 | 26.7% | 16.7% | 20.0% | 20.0% |
| 4 | 16.7% | 23.3% | 16.7% | 10.0% |
| 5 | 33.3% | 20.0% | 13.3% | 6.7% |
| 6 | 6.7% | 6.7% | 23.3% | 30.0% |
| 7 | - | 6.7% | - | 13.3% |
| 8 | 3.3% | - | 3.3% | - |
| 9 | - | 3.3% | - | 3.3% |
| Average Score | $4.0 \pm 1.51$ | $4.0 \pm 1.87$ | $4.0 \pm 1.80$ | $4.6 \pm 2.12$ |

The difference observed in the appearance parameter can be associated with the product's color. The traditional formulation had the expected appearance of traditional dulce de leche in comparison with the light formulation which presented a lighter color.

The flavor of the traditional formulation was described as "not too sweet and not too cloying", being also approved in the light product. The texture, however, was disapproved in the traditional product, whilst marked as positive points in the light formulation by a large part of the volunteers.

Both the color and texture of the light dulce de leche are related to the absence of biochemical processes that occur in high temperatures. The low percentage of sugar during manufacturing reduces both the Mallard reaction, responsible for the color, and the total solids increase, responsible for the high viscosity of the final product [20]. Whilst the correction with milk powder and thickeners led to a product with better rheological properties, the caramel coloring adjustment did not have the same effectiveness. The control of these aspects is an advantage in regulating the product until the desired characteristics are based on the volunteers' feedback.

The buying intention of a product is associated with acceptance, where good acceptability had a higher intention, as seen in the data obtained for both formulations in Figure 1 below.

When comparing the buying intentions of each product, the results show a preference for the traditional dulce de leche, 30.00% of the volunteers would buy this product often if it was commercialized, while 20.00% would always buy it.

Concerning the light formulation, only 6.70% of the participants would always buy the product if commercialized, with 23.30% stating that they would never buy it.

The medium acceptance for the light dulce de leche led to a low buying intention and, inversely, the traditional formulation, classified as "like it very much", showed more than 50.00% of the distribution between "would always buy" and "would buy very often".

The primary objective of this study was to create a product with commercial potential and substantial consumer acceptance, with a primary focus on target consumers with specific dietary restrictions.

The Brazilian consumer market subjected to sensory evaluation is familiar with the traditional sweet flavor characteristic of dulce de leche, which is inherently associated with a high caloric content. In contrast, other consumer markets, particularly those unaccustomed to such consumption, may exhibit a reduced comparative effect, thus rendering increased acceptance in foreign markets more likely and facilitating potential export opportunities. Moreover, products of this nature may find robust acceptance in regions and markets that value and require foods with elevated caloric and nutritional content.

The findings derived from the purchase intention survey align with the observations made during product elaboration, underscoring the capacity to optimize sensory attributes for the purpose of augmenting both acceptance and purchase intent.

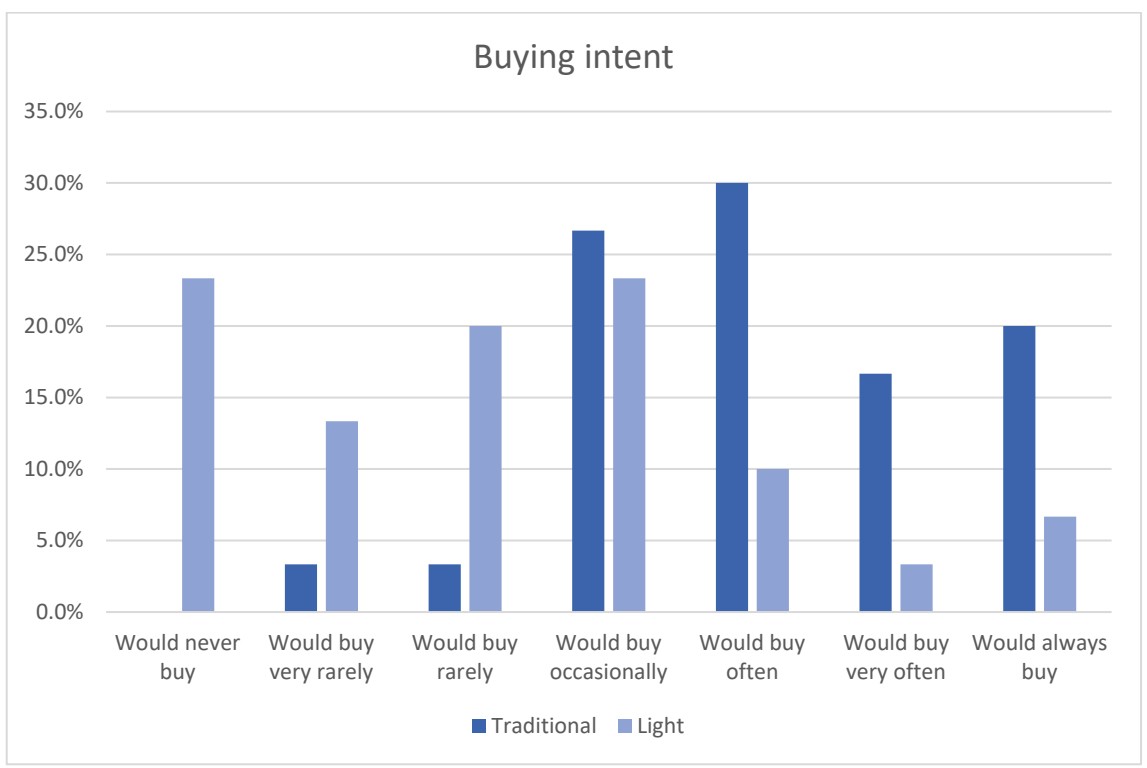

**Figure 1.** Traditional dulce de leche and light dulce de leche buying intent.

Subsequent studies may be conducted, using this work as a starting point, to assess variations in ingredient concentrations, including the exploration of alternative raw materials such as lactose-free milk powder and different sweeteners and thickeners. These explorations aim to enhance the functional properties of the product, particularly in terms of reducing its sugar and energy content. Additionally, the introduction of preservatives can be investigated to enable comparisons in physical-chemical and microbiological stability through extended quality control assessments.

**4. Conclusions**

Although the light formulation does not meet the energy reduction required by Brazilian legislation, the data show a promising feasibility with high commercial applicability after adjustments to the formulation. The physicochemical and microbiological parameters of the developed formulations proved to be adequate, contributing to the development of a product with good stability. Such properties corroborate the possibility of commercialization, especially in the international market, whose acceptance may be greater due to differences in taste preferences. The methodologies used proved to be promising for the development of new products with functional characteristics, serving as a standard for future developments aimed at improving product acceptance and purchase intention. This work can even be the basis for the development of different products in addition to dulce de leche, which aim to allow access to consumers with restrictions by maintaining high quality and low cost.

**Author Contributions:** Conceptualization, V.F.M., L.A.P. and F.B.N.K.; Methodology, V.F.M., L.A.P., F.B.N.K. and L.A.M.K.; Validation, V.F.M., L.A.P., F.B.N.K. and L.A.M.K.; Formal analysis, V.F.M. and L.A.M.K.; Investigation, V.F.M., L.A.P., F.B.N.K. and L.A.M.K.; Resources, V.F.M. and L.A.M.K.; Data curation, V.F.M., L.A.P. and F.B.N.K.; Writing—original draft preparation, V.F.M., L.A.P. and F.B.N.K.; Writing—review and editing, V.F.M., L.A.P., F.B.N.K. and L.A.M.K.; Supervision, V.F.M. and L.A.M.K.; Project administration, V.F.M., L.A.P., F.B.N.K. and L.A.M.K.; Funding acquisition, V.F.M. and L.A.M.K. All authors have read and agreed to the published version of the manuscript.

**Funding:** This research was funded by FAPERJ-Fundação de Amparo à Pesquisa do Estado do Rio de Janeiro, Process SEI 260003/002531/2021 and SEI 260003/003283/2022.

**Institutional Review Board Statement:** Ethical review and approval were waived for this study due to the data having been collected during an internal scientific fair organized within the Veterinary school with volunteers of legal age after completing the term of responsibility and explaining the intention of the project in order to obtain a preliminary result for future sensory studies.

**Informed Consent Statement:** Informed consent was obtained from all subjects involved in the study.

**Data Availability Statement:** The data presented in this study are available on request from the corresponding author.

**Acknowledgments:** The authors would like to thank to FAPERJ-Fundação de Amparo à Pesquisa do Estado do Rio de Janeiro for scholarship and funding. To the postgraduation course Higiene Veterinária e Tecnologia de Produtos de Origem Animal of the Federal Fluminense Unversity. the Departamento de Tecnologia de Alimentos at the same university (MTA-UFF), to the Mycology and Mycotoxin Laboratory at the Minas Gerais Federal University (LAMICO—UFMG).

**Conflicts of Interest:** The authors declare no conflict of interest.

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
