# Peer review of "Elaboration and Characterization of Dulce de Leche with Reduced Sugar Content"

_2624-862X, doi:10.3390/dairy4040043_

Round 1

Reviewer 1 Report

Comments and Suggestions for Authors

This article presents the noteworthy negative findings concerning light Dulce de Leche, which hold significance for the scientific community. I have some reservations about the project's design, and there is room for further refinement in this work.

1. The authors introduced additional raw materials when formulating the light Dulce de Leche, such as whole milk powder, which inherently contains sugar content.

2. Have the authors experimented with different proportions or alternative raw materials to reduce sugar, such as different sugar substitutes? I recommend exploring various raw material proportions and providing an explanation of the methodology used to arrive at the reported values.

3.    The unit conversion (kcal = kJ) is a fundamental concept and doesn't require citation.

4.    Inconsistencies exist in the paper regarding numeric representation. While the use of commas for numeric values is a common practice in Brazil, instances of employing the decimal system create readability issues and confusion.

5.  The authors should explain abbreviations like AOAC and AFM1 to ensure global accessibility and comprehension.

6.    A language review is advisable, as grammatical errors were detected in certain sections.

Comments on the Quality of English Language

The language is fine overall little improvement is needed in some sections. 

Reviewer 2 Report

Comments and Suggestions for Authors

My general opinion about the MN is positive and therefore it can be accepted with some modifications. I just have a series of considerations. The paper may be considered an useful contribute to a food quite common in South American country. Therefore, any information to change some characteristics is interesting.

Page 4 line 137 change Moebus et al in Moebus et al.

Page 11 line 377 the sentence promising for the development of new products with functional characteristics, explore an interesting development of the research. Please, clarify how the reported product should be apply the characteristics of a functional food and if there is an adequate market. In this regards, it is important that the authors have introduced the evaluation of the taste. The populations of Brazil are clearly used to the extremely sweet taste of this product, but lowering this aspect it could be possible also hypothesize exportation. Another aspect could be the best conditions of conservation for long time. Consider that this kind of product could be very useful for parts of Africa where there is an desperate need of foods rich in calories.

In conclusion, it should be necessary to interest also the readers which do not have direct interest or experience of this kind of product.

Several references could be added, like that of Carneiro et al 2021.

Round 2

Reviewer 1 Report

Comments and Suggestions for Authors

I have reviewed the revised manuscript and recommend the manuscript for publication in the present form.